# *LncBRM* initiates YAP1 signalling activation to drive self-renewal of liver cancer stem cells

Pingping Zhu[1,*], Yanying Wang[1,*], Jiayi Wu[1,2], Guanling Huang[1,2], Benyu Liu[1,2], Buqing Ye[1], Ying Du[1], Guangxia Gao[1,2], Yong Tian[3], Lei He[4] & Zusen Fan[1,2]

Liver cancer stem cells (CSCs) may contribute to the high rate of recurrence and heterogeneity of hepatocellular carcinoma (HCC). However, the biology of hepatic CSCs remains largely undefined. Through analysis of transcriptome microarray data, we identify a long noncoding RNA (lncRNA) called *lncBRM*, which is highly expressed in liver CSCs and HCC tumours. *LncBRM* is required for the self-renewal maintenance of liver CSCs and tumour initiation. In liver CSCs, *lncBRM* associates with BRM to initiate the BRG1/BRM switch and the BRG1-embedded BAF complex triggers activation of YAP1 signalling. Moreover, expression levels of *lncBRM* together with YAP1 signalling targets are positively correlated with tumour severity of HCC patients. Therefore, *lncBRM* and YAP1 signalling may serve as biomarkers for diagnosis and potential drug targets for HCC.

[1] Key Laboratory of Infection and Immunity of CAS, CAS Center for Excellence in Biomacromolecules, Institute of Biophysics, Chinese Academy of Sciences, 15 Datun Road, Chaoyang District, Beijing 100101, China. [2] University of Chinese Academy of Sciences, Beijing 100049, China. [3] Key Laboratory of RNA Biology and Beijing Noncoding RNA Laboratory, Institute of Biophysics, Chinese Academy of Sciences, Beijing 100101, China. [4] Department of Hepatobiliary Surgery, PLA General Hospital, Beijing 100853, China. * These authors contributed equally to this work. Correspondence and requests for materials should be addressed to Z.F. (email: fanz@moon.ibp.ac.cn).

Hepatocellular carcinoma (HCC) is the most prevalent subtype of liver cancer and ranks the third leading cause of cancer-related deaths[1]. Liver transplantation and surgical resection are the first-line treatment for HCC. Even after surgical resection, the 5-year survival rate of HCC patients remains poor, owing to high recurrence rates. The high rate of recurrence and heterogeneity are the two major features of HCC[2]. Cancer stem cells (CSCs) have been defined to be a small subset of cancer cells within the tumour bulk, exhibiting self-renewal and differentiation capacities[3]. CSCs may well contribute to tumour initiation, metastasis, recurrence, as well as drug resistance[3–5]. Liver CSCs can be enriched by some defined surface markers[6–8]. Several recent studies reported that Wnt/β-Catenin, Notch, Hedgehog, transforming growth factor-β, and phosphatase and tensin homologue signalling pathways are implicated in the regulation of liver CSC self-renewal[9–11]. However, the biology of liver CSCs remains largely elusive.

Long noncoding RNAs (lncRNAs) are transcripts longer than 200 nucleotides without protein-coding potentials[12]. Accumulating evidence shows that lncRNAs are involved in physiological and pathological progresses, including embryonic development, organ formation, X chromatin inactivation, tumorigenesis and so on refs 12–15. LncRNAs can recruit transcription factors and remodelling complexes to modulate gene expression[11] and they can also interact with messenger RNAs and regulate the stability of mRNAs. Several recent studies demonstrated that lncRNAs can associate with some important proteins and modulate their functions[16–18]. LncRNAs have been reported to be implicated in tumour formation and metastasis[16,17,19]. However, how lncRNAs regulate the self-renewal of liver CSCs remains largely unknown.

Yes-associated protein (Yap) and transcriptional co-activator with PDZ-binding domain motif (Taz) are transcriptional cofactors that shuttle between the cytoplasm to the nucleus where they interact with TEAD (TEA domain family member) transcription factors to activate downstream gene expression[20,21]. Accumulating evidence links the activity of Yap and Taz to tumorigenesis and chemoresistance[22–24]. However, how YAP1 signalling is activated in liver CSCs remains unknown.

Here we define a highly transcribed lncRNA in liver CSCs that we call lncBRM (lncRNA for association with Brahma (BRM), gene symbol LINCR-0003), which associates with BRM and modulates the BRG1/BRM switch in the BRG1-associated factor (BAF) complex, leading to activation of YAP1 signalling and promotion of liver CSC self-renewal.

## Results

**LncBRM is highly expressed in HCC tumours and liver CSCs.** Surface markers CD133 (ref. 25) and CD13 (ref. 6) have been widely used as liver CSC markers, respectively. We recently sorted a small subpopulation from HCC cell lines and HCC samples with these two combined makers and defined this subset of $CD13^+CD133^+$ cells as liver CSCs[11,25]. We performed transcriptome microarray analysis of $CD13^+CD133^+$ (liver CSCs) and $CD13^-CD133^-$ (non-CSCs) cells and identified 286 differentially expressed lncRNAs in liver CSCs compared with that in non-CSCs[11]. We previously showed that an uncharacterized lncRNA lncTCF7 regulates the maintenance of liver CSCs through recruitment of the SWI/SNF complex to activate Wnt signalling. Among the differentially expressed lncRNAs in liver CSCs, we chose top ten highly expressed lncRNAs and silenced these lncRNAs in HCC cell lines for in vitro oncosphere formation assays. We noticed that lncBRM depletion most dramatically inhibited oncosphere formation (Fig. 1a). This result was further validated by serial sphere

formation assays (Supplementary Fig. 1A,B). In addition, we deleted lncBRM in Hep3B and Huh7 cells by CRISPR/Cas9 technology and found that lncBRM knockout (KO) surely impaired serial sphere formation (Supplementary Fig. 1C,D). Notably, lncBRM knockdown did not affect the expression of its nearby genes (Supplementary Fig. 1E,F), suggesting that lncBRM exerts its function in trans.

LncBRM located on human chromosome 5 between the ACTBL2 and PLK2 genes (Supplementary Fig. 1G). LncBRM consisted of six exons, containing 1,321 nucleotides with a modestly conserved locus according to Phylop analysis (Supplementary Fig. 1G). The full length of lncBRM was further amplified by rapid-amplification of complementary DNA ends approaches and validated by sequencing (Supplementary Fig. 1H). In addition, lncBRM had no protein-coding potential (Supplementary Fig. 1I,J). LncBRM was highly expressed in HCC primary tumour tissues through northern blotting (Fig. 1b) and it was also highly transcribed in advanced HCC samples through quantitative reverse transcriptase–PCR (RT–qPCR; Fig. 1c), further validated by in situ hybridization (Fig. 1d). These results indicate that lncBRM is highly expressed in HCC tumour tissues.

Considering high expression levels of lncBRM in liver CSCs based on transcriptome data, we next wanted to confirm lncBRM expression in liver CSCs from HCC primary samples. We observed that lncBRM was indeed highly expressed in liver CSCs derived from six HCC primary samples (Fig. 1e). Of 33 primary samples we tested, we selected out these 6 HCC samples with high expression levels of lncBRM (1, 8, 9, 13, 17 and 20; Supplementary Table 1) and used these 6 HCC samples for our following studies. We then incubated HCC primary cells and cell lines for sphere formation, followed by enrichment of oncosphere cells (Sphere) and non-sphere cells (Non-sphere) for further examination. We found that lncBRM was also highly expressed in oncosphere cells derived from HCC primary samples and cell lines (Fig. 1e,f), which was further validated by RNA fluorescence in situ hybridization (RNA-FISH; Fig. 1g). In addition, lncBRM was mainly located in the nucleus (Fig. 1g). Similar observations were obtained by nucleocytoplasmic fractionation of HCC cells (Fig. 1h). Altogether, lncBRM is highly expressed in HCC tumour tissues and liver CSCs.

**LncBRM is required for the self-renewal of liver CSCs.** We next wanted to determine the role of lncBRM in the self-renewal maintenance of liver CSCs. We silenced lncBRM in 6 HCC primary tumour cells (Fig. 2a and Supplementary Fig. 2A), followed by sphere formation assays. For short hairpin RNA (shRNA) designs, we searched for 12 shRNA candidates against lncBRM and selected out 2 most efficient silencing shRNAs (termed shLncBRM1 and shLncBRM2) for our following studies. We noticed that lncBRM depletion markedly repressed sphere formation (Fig. 2a). Reduced self-renewal capacity was further verified in lncBRM-silenced cells through serial sphere formation assays (Fig. 2b). We established stably silenced lncBRM and scrambled RNA (shCtrl)-treated HCC primary tumour cells and injected $1 \times 10^6$ lncBRM-silenced and shCtrl cells into BALB/c nude mice, followed by measurement of tumour volumes every 4 days. LncBRM-depleted cells significantly reduced tumour propagation compared with shCtrl-treated cells (Fig. 2c), which was further validated by in vivo serial xenograft passage assays (Fig. 2d). Furthermore, lncBRM silencing displayed much weaker tumour initiation and tumorigenic cell frequency as measured by a limiting dilution xenograft analysis (Fig. 2e and Supplementary Table 2A). To validate the results by lncBRM knockdown, we also established lncBRM KO cells by using a CRISPR/Cas9 approach. LncBRM KO displayed impaired self-renewal capacities of liver

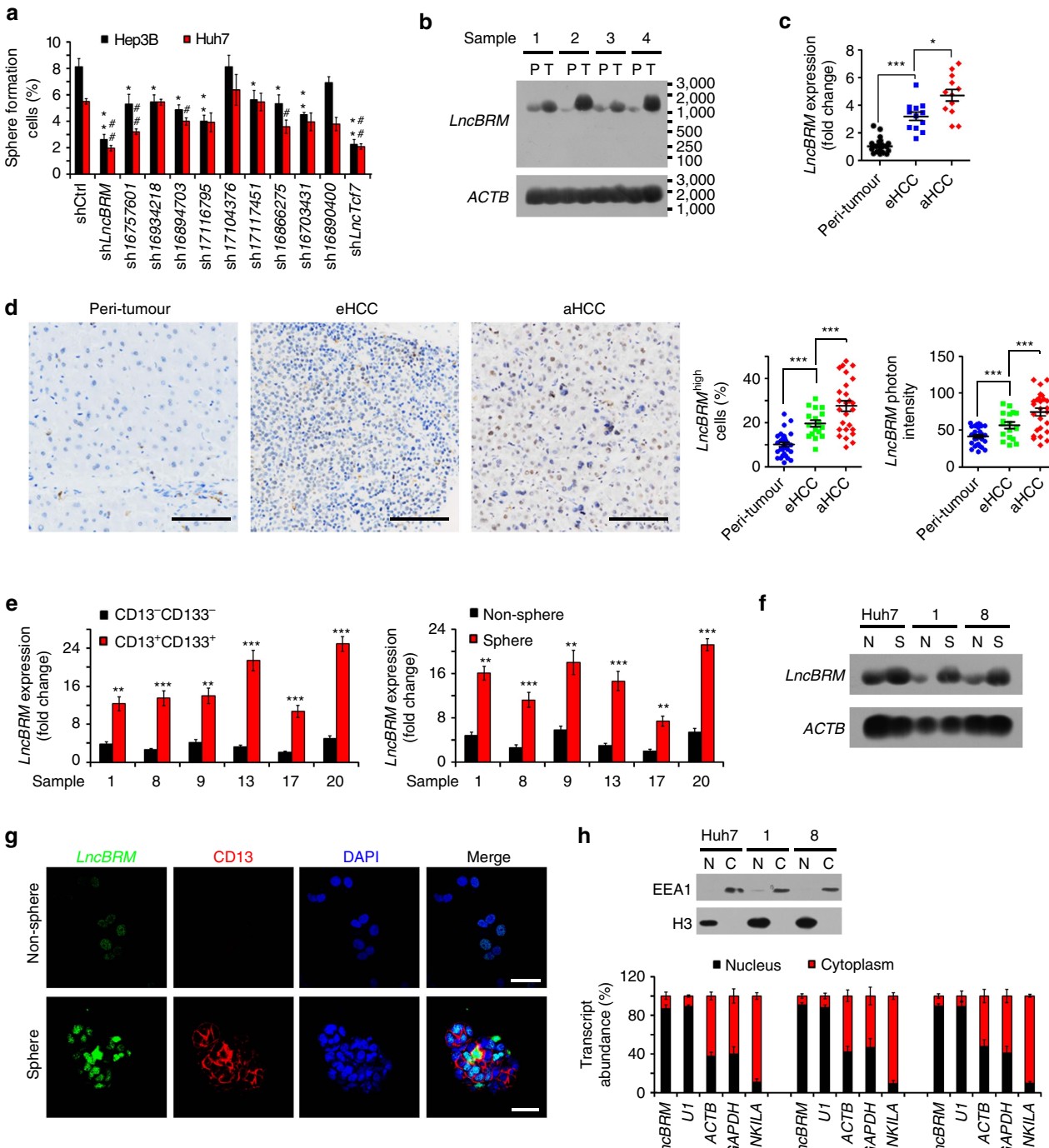

**Figure 1 | LncBRM is highly expressed in HCC tumours and liver CSCs.** (**a**) The indicated lncRNAs were silenced using pSiCoR lentivirus, followed by sphere formation assays. *, **, for Hep3B cells, Hep3B shlncBRM versus Hep3B shCtrl; #, ##, for Huh7 cells, Huh7 shlncBRM versus Huh7 shCtrl. (**b**) Total RNAs were extracted from peri-tumour (P) and tumour (T) tissues, followed by northern blotting. *ACTB* served as a loading control. (**c**) Primary HCC samples were prepared for examination of *lncBRM* expression using RT–qPCR. aHCC, advanced HCC; eHCC, early HCC. (**d**) *LncBRM* was detected by *in situ* hybridization. *LncBRM* highly expressed cells (middle panel) and *lncBRM* photon intensity (right panel) were calculated by Image-Pro Plus 6 and shown as scatter plot (means ± s.e.m.). Scale bars, 100 μm. (**e**) Liver CSCs (CD13⁺CD133⁺) and non-CSCs (CD13⁻CD133⁻) were sorted from HCC samples, followed by detection of *lncBRM* using RT–qPCR (left panel). Oncospheres and non-spheres derived from HCC primary tumour cells were analysed similarly. Expression levels of *lncBRM* were normalized to that of non-tumour sample 17 as a baseline level. (**f**) *lncBRM* was examined in oncospheres and non-spheres with northern blotting. N, non-sphere; S, sphere. (**g**) Non-spheres and spheres were stained with *lncBRM* probes and CD13 antibody for confocal microscopy. Scale bars, 20 μm. (**h**) Nucleocytoplasmic fractionation of oncosphere cells was performed and followed by immunoblotting (upper panel) and RT–qPCR (lower panel). U1 RNA served as a nuclear location control and *NKILA* was used as a cytoplasmic location control. Data are shown as means ± s.d. Two tailed Student's *t*-test was used for statistical analysis; *P < 0.05, **P < 0.01, ***P < 0.001; #P < 0.05, ##P < 0.01. Data are representative of at least three independent experiments.

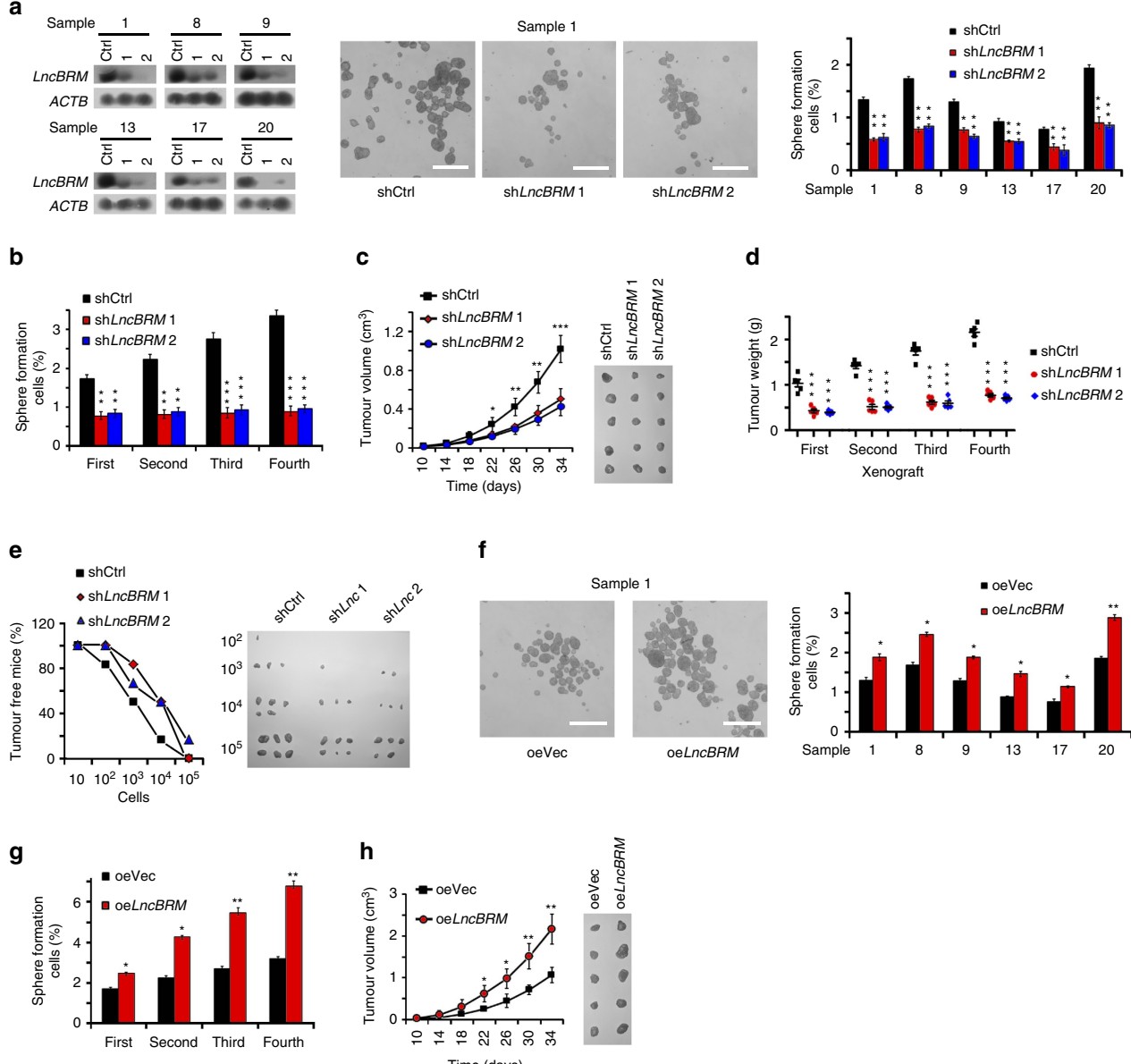

**Figure 2 | LncBRM is required for the self-renewal maintenance of liver CSCs.** (**a**) LncBRM-silenced cells from HCC samples were established and examined by northern blotting (left panel), followed by sphere formation assays. Representative sphere images were shown in the middle panel and statistic ratios were shown in the right panel. (**b**) Declined self-renewal capacities of lncBRM-depleted cells were detected by serial sphere formation assays. (**c**) LncBRM-depleted and control cells ($1 \times 10^6$) were injected into BALB/c nude mice, followed by measurement of tumour volumes. $n = 5$ for each group. (**d**) LncBRM-silenced primary cells were established with pSiCoR lentivirus and used for subcutaneous injection. Xenograft tumours were used for serial tumour implantation. $n = 5$ for each group. (**e**) LncBrm silenced and shCtrl cells (10, $10^2$, $10^3$, $10^4$ and $10^5$) were subcutaneously injected into BALB/c nude mice for observation of tumour growth. Tumour-free mice ratios and tumour sizes are shown. (**f**) LncBRM-overexpressing primary cells were established and used for sphere formation assays. (**g**) LncBRM-overexpressing cells were examined by serial sphere formation assays. (**h**) LncBRM-overexpressing cells were injected into BALB/c nude mice for observation of tumour growth and measurement of tumour volume every 4 days. $n = 5$ for each group. Scale bars, 500 μm (**a,f**). Data are shown as means ± s.d. Two tailed Student's t-test was used for statistical analysis; *$P < 0.05$, **$P < 0.01$ and ***$P < 0.001$. Data are representative of four independent experiments.

CSCs (Supplementary Fig. 2B,C), which were really in agreement with those of lncBRM knockdown results. Therefore, these results indicate that lncBRM depletion abrogates the stemness of liver CSCs.

We then generated lncBRM stably overexpressed HCC primary tumour cells using lentivirus (Supplementary Fig. 2D), followed by examination of sphere formation and self-renewal capacities. We observed that lncBRM overexpression augmented in vitro oncosphere formation and self-renewal (Fig. 2f,g), as well as

promoted xenograft tumour propagation and tumorigenic cell frequency (Fig. 2h, Supplementary Fig. 2E,F and Supplementary Table 2B). Collectively, lncBRM promotes the self-renewal of liver CSCs and in vivo tumour propagation.

**LncBRM modulates the BRG1/BRM switch.** To determine associated proteins of lncBRM, we performed RNA pulldown assays in oncosphere cell lysates with biotin-labelled

*lncBRM*. One overtly differential band around 170 kDa appeared by silver staining (Fig. 3a) and identified to be BRM by mass spectrometry (Supplementary Fig. 3A). In fact, *lncBRM* could precipitate BRM in liver oncosphere cell lysates by RNA immunoprecipitation, but not other BAF complex components

(Fig. 3b and Supplementary Fig. 3B). Through domain mapping, we defined that 3 segment (607~951) of *lncBRM* interacted with BRM (Fig. 3c). In addition, stable stem-loop structure of the binding fragment was predicted by RNA folding analysis (Supplementary Fig. 3C). The interaction of *lncBRM* with BRM

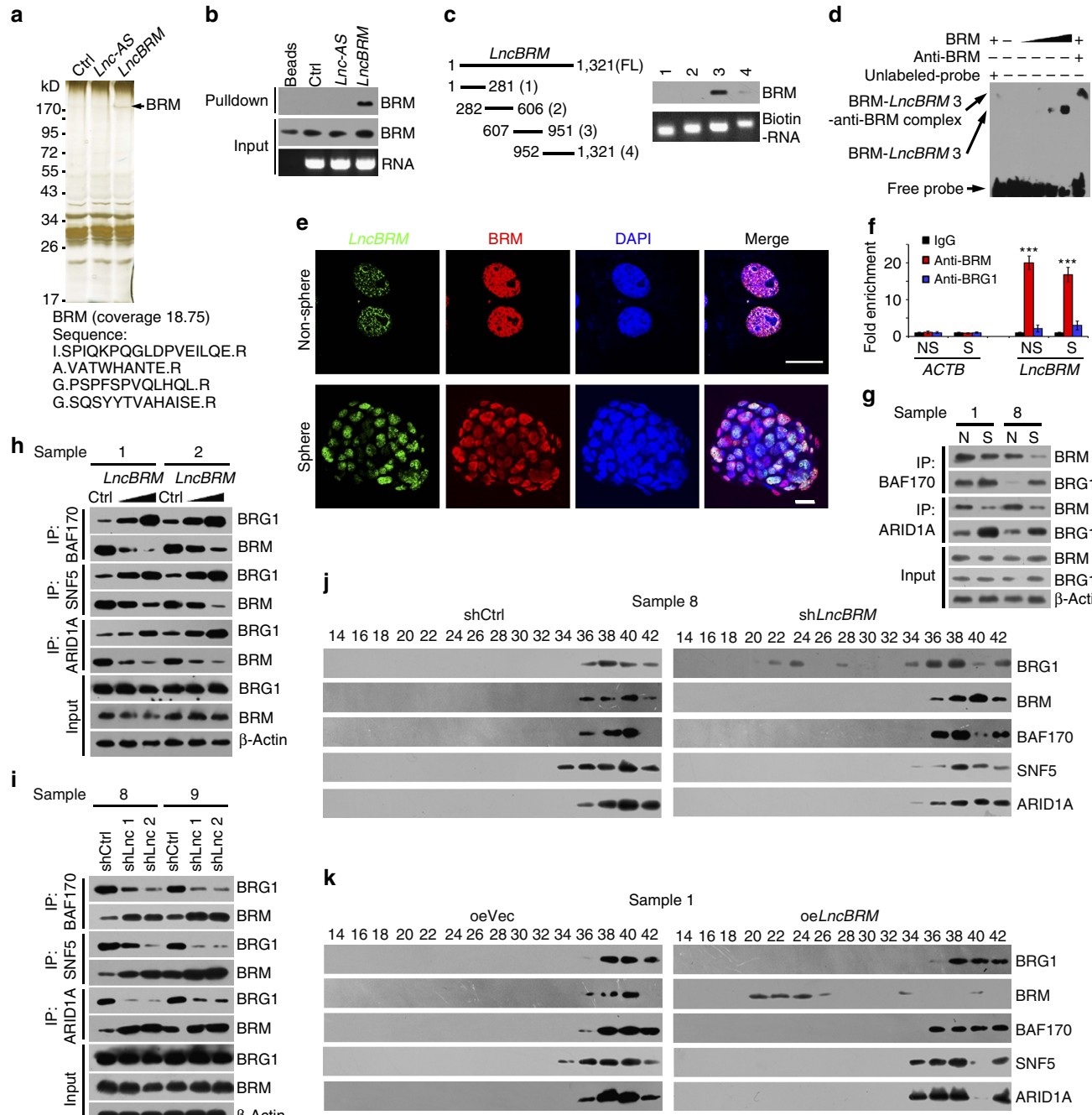

**Figure 3 | *LncBRM* associates with BRM to initiate the BRG1/BRM switch.** (**a**) *LncBRM* intron sequence (Ctrl), *lncBRM* antisense (*Lnc-AS*) and *lncBRM* transcripts were labelled with biotin and incubated with oncosphere lysates, followed by silver staining and mass spectrometry. Black arrow denotes BRM. (**b**) RNA pulldown was conducted using *lncBRM* transcript, followed by immunoblotting. (**c**) Domain mapping of *lncBRM* transcript. (**d**) *LncBRM* was incubated with increased doses of BRM, followed by electrophoretic mobility shift assay (EMSA). The 3 segment of *lncBRM* was labelled with biotin for probing. (**e**) Non-spheres and spheres were visualized by fluorescence *in situ* hybridization (FISH). Scale bar, 10 μm. (**f**) Antibodies against BRM or BRG1 were used for RNA immunoprecipitation, followed by RT–qPCR. *ACTB* served as a negative control. (**g**) Spheres (S) and non-sphere cells (N) were lysed and followed by immunoprecipitation with BAF170 and ARID1A antibodies. BRG1 and BRM enrichment was analysed with western blotting. (**h**) Different doses of *lncBRM* transcripts were incubated with oncosphere lysates and followed by co-immunoprecipitation (co-IP). (**i**) *LncBRM*-depleted HCC primary spheres were lysed for co-IP as in **h**. (**j,k**) The indicated oncosphere lysates were fractionated and followed by size fractionation with glycerol gradient ultracentrifugation. Elute gradients were used for western blotting. Data are shown as means ± s.d. Two tailed Student's *t*-test was used for statistical analysis, ***P < 0.001. Data represent at least three independent experiments.

was further validated by an RNA electrical mobility shift assay (EMSA; Fig. 3d). The co-localization of *lncBRM* and BRM in non-sphere and oncosphere cells was visualized by immuno-fluorescence staining (Fig. 3e) and mainly distributed in the nucleus. These data indicate that *lncBRM* associates with BRM in the nuclei of liver CSCs.

The SWI/SNF complex is an evolutionally conserved multi-subunit chromatin remodelling complex, which uses ATPase subunit BRM or BRG1 to provide ATP for remodelling nucleosomes[26,27]. Mammals contain two yeast BRM homologues BRM and BRG1, which are widely expressed and amino acid sequences are 75% identical. Nonetheless, these subunits are mutually exclusive, as each SWI/SNF complex possesses BRM or BRG1, forming a BRM-embedded or BRG1-embedded BAF complex. However, the physiological roles of the BRM-embedded or BRG1-embedded BAF complex have not been defined yet. We noticed that anti-BRM antibody but not anti-BRG1 antibody could precipitate *lncBRM* in oncosphere cell lysates (Fig. 3f), validating the interaction between BRM and *lncBRM*. We observed that BRM-embedded BAF complex was dramatically declined in oncosphere cells (Fig. 3g), but BRG1-embedded BAF complex predominantly existed in these cells, suggesting a BRG1/BRM switch in liver CSCs. The anti-BRM (catalogue number 11966) and anti-BRG1 (catalogue number 21634-1-AP) were specifically recognized their respective proteins.

We then incubated *lncBRM* transcripts with oncosphere lysates for co-immunoprecipitation assays. We observed that *lncBRM* increased assembled BRG1-embedded BAF complex (Fig. 3h), but not BRM-embedded complex. Moreover, *lncBRM* depletion impaired the BRG1-embedded BAF complex in oncosphere cells (Fig. 3i). To further verify this result, we used *lncBRM*-silenced sphere lysates to conduct size fractionation assays as previously described[25]. BRG1 and BRM constitute the BRG1- and BRM-embedded BAF complex, respectively, which can be detected by sucrose density gradient centrifugation. We noticed that lncBRM depletion indeed caused more free BRG1 in oncospheres (Fig. 3j), suggesting impaired assembly of the BRG1-biased BAF complex. In contrast, *lncBRM* overexpression led to more free BRM in oncospheres (Fig. 3k), suggesting impaired assembly of the BRM-biased BAF complex. Finally, we found that *lncBRM* depletion did not significantly impact BRM stability or ATPase activity (Supplementary Fig. 3D,E). Collectively, *lncBRM* associates with BRM to sequester BRM away, to initiate assembly of BRG1-biased BAF complex in liver CSCs, resulting in the BRG1/BRM switch.

**BRG1 initiates activation of YAP1 signalling in liver CSCs.** We next wanted to explore the function of BRG1-embedded BAF complex in the self-renewal maintenance of liver CSCs. We generated *BRG1* and *BRM* KO cells using CRISPR/Cas9 approaches as previously described[25]. We detected expression levels of major self-renewal-related pathways in *BRG1* KO and *BRM* KO spheres. We found that *BRG1* deletion dramatically abrogated YAP1 signalling (Fig. 4a). By contrast, *BRM* KO remarkably activated YAP1 signalling. These results were further validated by immunoblotting (Fig. 4b), suggesting that BRG1-embedded BAF complex could be involved in the activation of YAP1 signalling.

Through domain mapping, we observed that BRG1 bound to *YAP1* promoter (Fig. 4c) and validated by luciferase assays (Fig. 4d). However, BRM had no such binding activity. The interaction of BRG1 with *YAP1* promoter was further verified by EMSA assays (Fig. 4e). We noticed that BRG1 deletion significantly suppressed *YAP1* promoter activation by DNase I sensitivity assays (Fig. 4f). By contrast, *BRM* deletion enhanced

*YAP1* promoter activation. In parallel, H3K4me3 antibody did not enriched at *YAP1* promoter regions in *BRG1* KO oncospheres (Fig. 4g). By contrast, it could precipitate the *YAP1* promoter regions in *BRM* KO cells, suggesting the activation of *YAP1* promoter. We next deleted the BRG1-binding region of *YAP1* promoter (*YAP1P*KO) using a CRISPR/Cas9 approach. In *YAP1P*KO cells, BRG1 depletion has no effect on activation of YAP1 signalling (Fig. 4h). However, BRG1 depletion did suppress the YAP1 signalling in wild-type cells. These data indicate that BRG1 induces the activation of YAP1 signalling via binding its promoter.

**BRG1 drives YAP1 expression through a KLF4-dependent manner.** To further determine how YAP1 expression was activated by BRG1-embedded BAF complex, we analysed the BRG1-binding regions of *YAP1* promoter. We noticed that the BRG1-binding region of *YAP1* promoter contained a specific KLF4 binding sequence. KLF4, together with other three Yamanaka pluripotency factors, is able to reprogramme adult fibroblasts into induced pluripotent cells[28,29]. KLF4 can inhibit somatic genes in an early phase and subsequently initiate pluripotency genes in a late phase during cellular reprogramming[30,31]. Of note, KLF4 interacted with BRG1 but not BRM in oncospheres (Fig. 5a). Furthermore, KLF4 bound to the same BRG1-binding region of *YAP1* promoter (Fig. 5b), which was verified by EMSA assays (Fig. 5c). We next performed chromatin immunoprecipitation (ChIP) with anti-KLF4 antibody through oncosphere lysates, followed by size fractionation with detection by immunoblotting. We observed that KLF4 co-eluted with BRG1-embedded BAF complex components and Yap1 promoter (Fig. 5d), suggesting KLF4 recruits the BRG1-embedded BAF complex to Yap1 promoter. In addition, KLF4 deletion abolished the binding of BRG1 with *YAP1* promoter (Fig. 5e) and subsequently impaired *YAP1* transcription (Fig. 5f). Importantly, KLF4 rescue in KLF4-depleted HCC cells could restore *YAP1* activation (Supplementary Fig. 4A). These results indicate that KLF4 is involved in the *YAP1* activation.

We next established *YAP1* promoter mutant (Mut) cells by replacement of the *KLF4*-binding region of *YAP1* promoter with a CRISPR/Cas9 approach. We observed that the *YAP1* promoter mutant abrogated the interaction of KLF4 with *YAP1* promoter (Supplementary Fig. 4B). H3K4me3 and H3K27Ac enrichment at gene promoters denotes the activation of genes. Consistently, the *YAP1* promoter mutant reduced enrichment of BRG1 and H3K4me3 (Supplementary Fig. 4B). Moreover, overexpression of BRG1 in wild-type cells promoted *YAP1* promoter activation, whereas BRG1 overexpression in *YAP1* promoter mutant cells failed to initiate *YAP1* promoter activation (Fig. 5g). In parallel, overexpression of *lncBRM* achieved similar results (Fig. 5h). We also showed that lncBRM recruited the BRG1-embedded BAF complex to *YAP1* promoter (Supplementary Fig. 4C,D). These observations were further validated by luciferase assay (Supplementary Fig. 4E), ChIP assay (Supplementary Fig. 4F) and western blotting (Supplementary Fig. 4G). These data indicate that *lncBRM* and BRG1 induces *YAP1* expression in a KLF4-dependent manner. Collectively, *YAP1* promoter enriches KLF4 and KLF4 recruits the BRG1-embedded BAF complex, to initiate the activation of *YAP1* transcription.

**YAP1 is required for the self-renewal of liver CSCs.** We further determined the physiological role of *YAP1* signalling in the regulation of liver CSCs. Of note, YAP1 overexpression in non-sphere cells significantly restored sphere formation and enhanced self-renewal potential by serial passage assays (Fig. 6a). Conversely, the YAP1-specific inhibitor Verteporfin treatment in

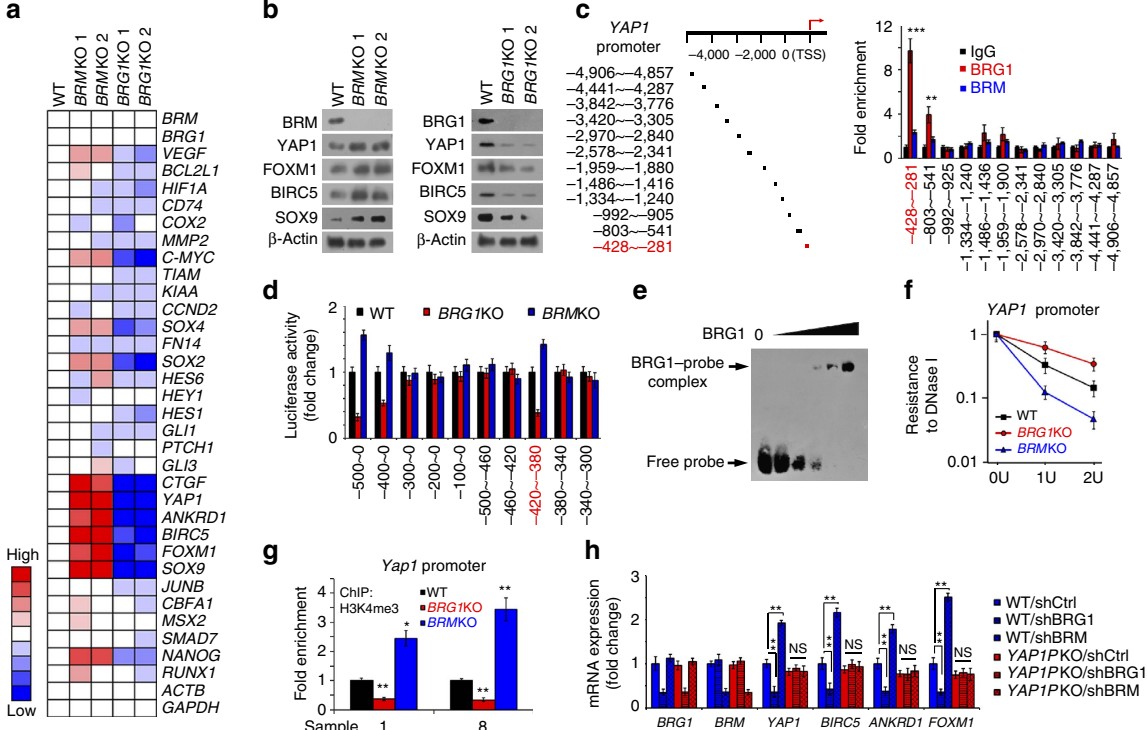

**Figure 4 | BRG1-embedded BAF complex initiates YAP1 signalling in liver CSCs.** (**a**) *BRG1* and *BRM* KO cells were established by CRISPR/Cas9 approaches, followed by examination of main self-renewal pathway target genes. Gene fold changes were determined by RT–qPCR. CRISPR/Cas9 caused frameshift mutations with no changes in mRNA levels. (**b**) YAP1 targets were tested by immunoblotting in *BRM* KO spheres (left panel) or *BRG1* KO spheres (right panel). (**c**) Schematic diagram of *YAP1* promoter (left panel) and domain mapping (right panel). HCC primary spheres were collected for ChIP assay with BRG1, BRM antibodies. TSS, transcription start site. (**d**) The binding region of *YAP1* promoter to BRG1 was validated by luciferase assay. (**e**) Biotin-labelled *YAP1* promoter region ($-420 \sim -380$ bp) was used for EMSA assay. BRG1 was immunoprecipitated from Huh7 spheres using BRG1-specific antibody. (**f**) Oncosphere nuclear lysates of the indicated cells were treated with DNase I, followed by real-time PCR. (**g**) *BRG1* KO or *BRM* KO spheres were used for ChIP assays using H3K4me3 antibody. (**h**) BRG1-binding region of *YAP1* promoter was deleted in HCC primary tumour cells using CRISPR/Cas9 technology, followed by depletion of *BRG1* or *BRM*. Total RNA was extracted for PCR assay. YAP1PKO, *YAP1* promoter KO. Data are shown as means ± s.d. Two tailed Student's *t*-test was used for statistical analysis; $*P < 0.05$ and $**P < 0.01$; NS, not significant. Data are representative of three independent experiments.

non-sphere cells abrogated sphere formation and impaired self-renewal capacity (Fig. 6a). Consequently, YAP1 over-expression in non-CSC cells augmented expression levels of stemness surface markers and pluripotency factors (Supplementary Fig. 5A). By contrast, Verteporfin treatment showed opposite results. Therefore, YAP1 signalling is required for the self-renewal maintenance of liver CSCs.

We next deleted *YAP1* gene in HCC primary tumour cells using CRISPR/Cas9 technology (Supplementary Fig. 5B). We found that *YAP1* KO displayed impaired sphere formation (Fig. 6b) and reduced tumour-initiating capacity (Fig. 6c and Supplementary Table 2C). Consequently, *YAP1*-deficient cells decreased xenograft propagation (Fig. 6d). Conversely, YAP1-overexpressing cells enhanced sphere formation and xenograft propagation (Supplementary Fig. 5C and data not shown). Thus, YAP1 is essential for the self-renewal maintenance of liver CSCs and tumour initiation.

We next examined the role of YAP1 signalling target genes in liver CSCs. YM155 has been reported to be an inhibitor of BIRC5 expression[32], whereas Siomycin A is a specific inhibitor against FOXM1. Of note, YM155 or Siomycin A treatment significantly suppressed oncosphere formation (Fig. 6e). Similarly, BIRC5 or FOXM1 depletion also reduced sphere formation (Supplementary Fig. 5D,E), indicating YAP1 signalling is required for promoting self-renewal of liver CSCs. Finally, Verteporfin treatment abolished enhanced sphere formation induced by *lncBRM*

overexpression (Supplementary Fig. 5F). Consistently, YAP1 could rescue sphere formation reduced by *lncBRM* knockdown (Fig. 6f). Altogether, *lncBRM*-mediated YAP1 signalling is required for the self-renewal of liver CSCs and tumour propagation.

**BRG1 and YAP1 targets are positively related to HCC severity.** We analysed the expression levels of BRG1/BRM and Yap signalling target genes using online available data sets. We noticed that *BRG1* was highly expressed in HCC tumours (Fig. 7a), whereas *BRM* was lowly expressed in HCC tumour tissues derived from Wang's cohort (GSE14520; ref. 33). In addition, *YAP1* signalling target genes were also highly expressed in HCC tumours (Fig. 7b). Moreover, increased *BRG1* levels and decreased *BRM* levels were positively correlated with severity of HCC patients (GSE14520; Fig. 7c). In parallel, expression levels of YAP1 signalling targets were also related to HCC severity (Fig. 7d). Of note, increased *BRG1* levels and decreased *BRM* levels were also positively related to metastasis of HCC patients (Fig. 7e) and expression levels of YAP1 targets showed the same trend as BRG1 (Fig. 7f). We validated these observations by examination of HCC samples with RT–qPCR (Fig. 7g,h), immunoblotting (Fig. 7i) and immunohistochemical staining (Fig. 7j). Similar observations were also achieved by analysis of other two cohorts (Zhang's cohort (GSE25097) and Wang's

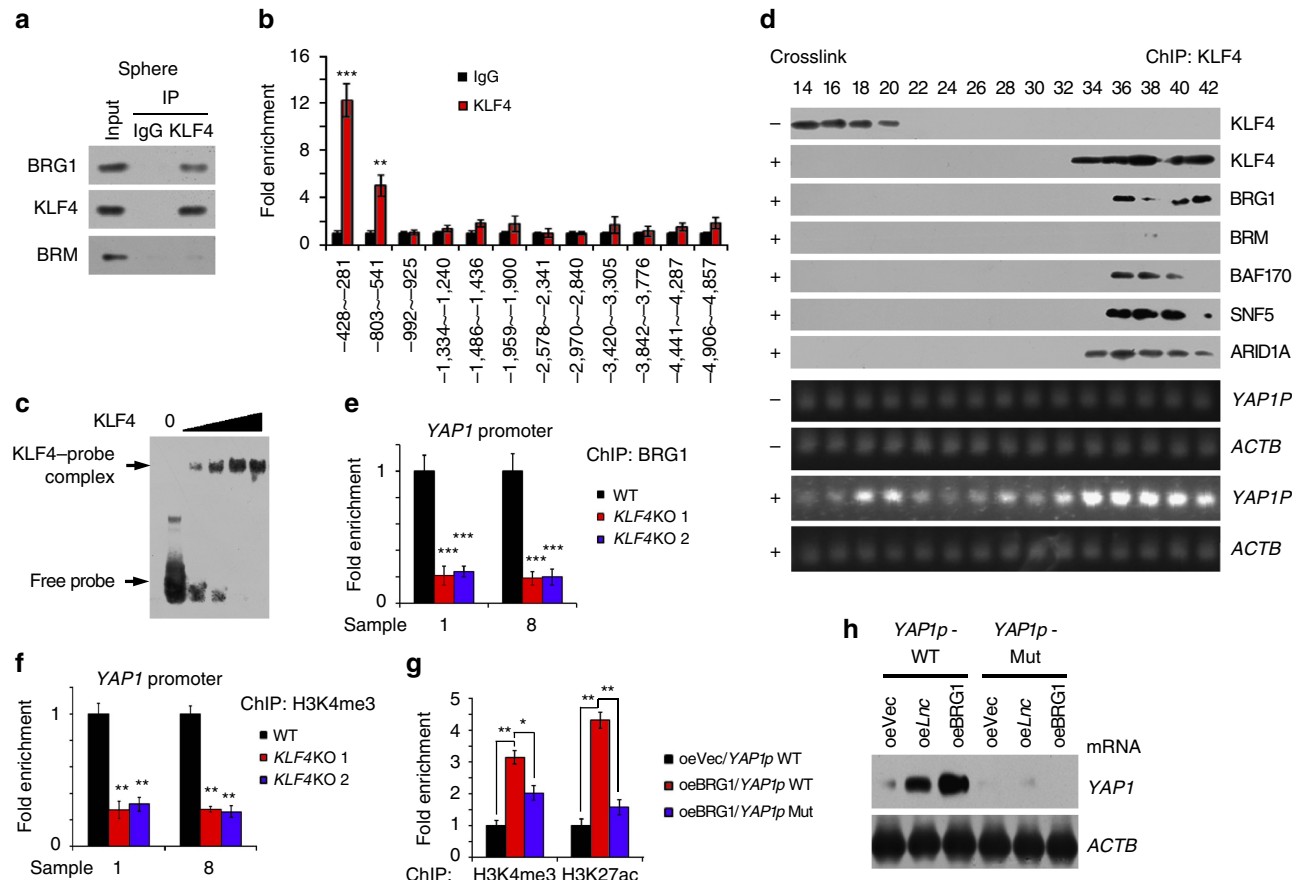

**Figure 5 | KLF4 binds *YAP1* promoter and recruits the BRG1-embedded BAF complex to initiate YAP1 expression.** (**a**) HCC primary spheres were collected for co-IP assays with KLF4 antibody. (**b**) ChIP assay was performed using Klf4 antibody. (**c**) The interaction of KLF4 with *YAP1* promoter was verified by EMSA assay. (**d**) HCC primary sphere cells were crosslinked with formaldehyde for ChIP assay with KLF4 antibody, followed by glycerol gradient ultracentrifugation. Elution gradients were concentrated for western blotting (upper panels) and PCR (lower panels) analyses. (**e,f**) *KLF4* KO cells were established using CRISPR/Cas9 technology and allowed for sphere formation, followed by ChIP assay using BRG1 (**e**) and H3K4me3 (**f**) antibodies. (**g**) BRG1 was overexpressed in *YAP1* promoter mutant (*YAP1p* Mut) and wild-type (WT) cells, followed by sphere formation. Oncosphere cells were used for ChIP assays with H3K4me3 and H3K27ac antibodies, followed by examination for *YAP1* promoter enrichment with real-time PCR. (**h**) *LncBRM* or BRG1 was overexpressed in *YAP1* promoter mutant and WT cells and collected for RNA extraction. *YAP1* mRNA expression was detected with northern blotting. *ACTB* served as a loading control. Data are shown as means ± s.d. Two-tailed Student's *t*-test was used for statistical analysis; *$P<0.05$, **$P<0.01$ and ***$P<0.001$. Data are representative of three independent experiments.

cohort (GSE54238); Supplementary Fig. 6A–E). However, for tumour stage, metastasis and survival analyses, Wang's cohort (GSE54238) and Zhang's cohort (GSE25097) were not suitable for these analyses because of limited clinical information (GSE25097) or sample numbers (GSE54238). In addition, several Yap1 target genes we focused on (*Ankrd1*, *Birc5* and so on) were not available in Wang's cohort (GSE54238) and Zhang's cohort (GSE25097) due to microarray platforms. More importantly, expression levels of YAP1 target genes were consistent with clinical prognosis of HCC patients (GSE14520; Supplementary Fig. 6F). Collectively, expression levels of BRG1 and YAP1 target genes are well related to cancer severity and prognosis of HCC patients.

## Discussion
CSCs may contribute to cancer relapse and metastasis due to their invasive and drug-resistant capacities[34]. Recently, several surface markers have been identified to enrich liver CSCs[6,8,11], whose heterogeneous markers may represent different cellular origins. However, the biology of liver CSCs remains largely unknown. We previously isolated a rare subset of CD13⁺CD133⁺ cells from most HCC cell lines and primary samples[11,25], and showed that the CD13⁺CD133⁺ subpopulation harbours robust self-renewal

and differentiation abilities, serving as liver CSCs. Based on our transcriptome microarray analysis of liver CSCs, we identified an uncharacterized lncRNA termed *lncBRM*, which maintains the stemness of liver CSCs in *trans*. *LncBRM* associates with BRM to initiate the BRG1/BRM switch and the BRG1-embedded BAF complex triggers activation of YAP1 signalling.

LncRNAs, as a new class, exhibit a wide range of expression levels and distinct cellular localizations, reflecting a large and diverse class of regulators[12,35]. LncRNAs can exert their functions through diverse modes, including cotranscriptional regulation, modulation of gene expression, scaffolding of nuclear or cytoplasmic complexes, as well as pairing with other RNAs[36]. Collectively, lncRNAs can function in *cis* to regulate expression of neighbouring genes or in *trans* to carry out many roles by various modes[35]. Several recent studies reported that lncRNAs are implicated in tumour initiation, invasion and metastasis[16,17,19,37,38]. However, the exact role of lncRNAs in liver CSCs remains largely undefined. Here we showed that *lncBRM* is highly expressed in HCC tumours and liver CSCs. *LncBRM* associates with BRM to make the BRG1/BRM switch and BRG1-embedded BAF complex initiates the activation of YAP1 signalling to sustain the self-renewal of liver CSCs. Of note,

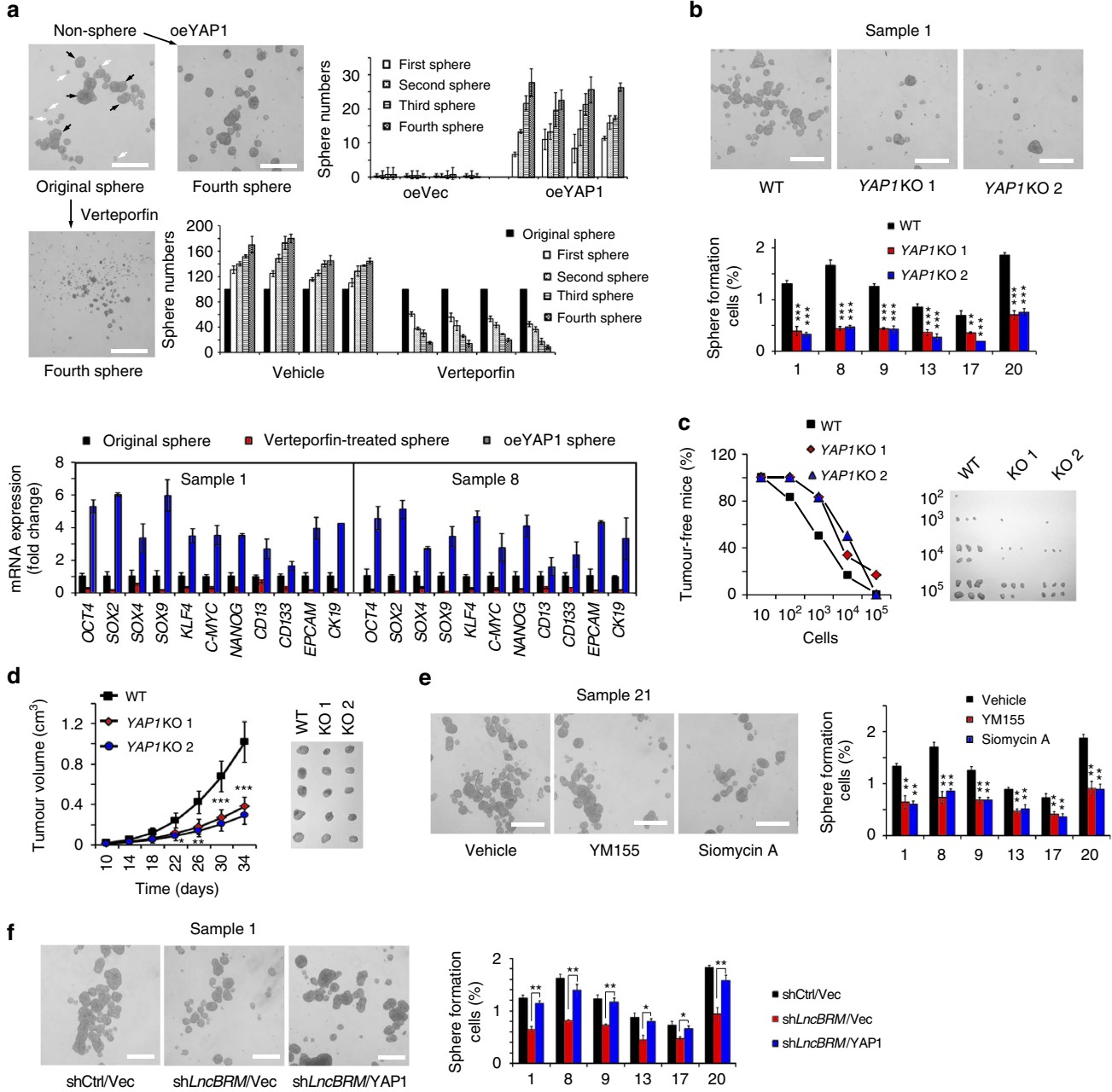

**Figure 6 | *LncBRM*-mediated YAP1 signalling is required for self-renewal of liver CSCs and tumour propagation.** (**a**) YAP1 was activated (overexpression) or inactivated (Verteporfin treatment) in non-sphere cells or sphere cells, followed by measurement of self-renewal by serial sphere formation assays. Fourth passage spheres and sphere numbers were shown (upper two panels). Two weeks later, original spheres, Verteportin-treated spheres and YAP1-overexpressing spheres were collected for mRNA detection (lower panels). (**b**) *YAP1* KO cells (*YAP1*KO) were used for sphere formation assays. (**c**) Wild-type (WT) and *Yap1*-deficient cells (10, $10^2$, $10^3$, $10^4$ and $10^5$) were injected into BALB/c nude mice for measurement of tumour formation. Tumour-free mice and established tumours were shown. $n = 6$ for each group. (**d**) $1 \times 10^6$ *YAP1* KO and control cells were injected into BALB/c nude mice for tumour growth. Tumour volume was calculated every 4 days. $n = 5$ for each group. (**e**) HCC primary tumour cells were treated with YM155 and Siomycin A for sphere formation assay. (**f**) YAP1 was rescued in *lncBRM*-depleted cells using pBPLV lentivirus and followed by sphere formation assay. Scale bars, 500 μm (**a**,**b**,**e**,**f**). Data are shown as means ± s.d.; *$P < 0.05$, **$P < 0.01$ and ***$P < 0.001$. Data are representative of four independent experiments.

*lncBRM* deletion failed to impact asymmetric division ratios and differentiation tendency. Surely, *lncBRM* and *lncTCF7* had their unique predicted tertiary structures. Of note, lncBRM KO did not affect the expression levels of *lncTCF7* and its target gene *TCF7* in liver CSCs (Supplementary Fig. 7A). Conversely, *lncTCF7* deletion did not have an impact on the expression levels of *lncBRM* and YAP1 either (Supplementary Fig. 7B). These data suggest that *lncBRM* and *lncTCF7* regulate the stemness of liver CSCs by using different pathways. In addition, we observed

that depletion of both *lncTCF7* and *lncBRM* more significantly impaired oncosphere formation than *lncTCF7* or *lncBRM* depletion alone (Supplementary Fig. 7C), suggesting that targeting both *lncTCF7* and *lncBRM* may be better than *lncBRM* or *lncTCF7* alone for potential therapy of HCC patients. Liver CSCs are really heterogeneous and many different types of liver CSCs have been reported by the expression of different CSC markers such as CD90, EpCAM, CD24, CD44 and so on. We found that *lncBRM* was also highly

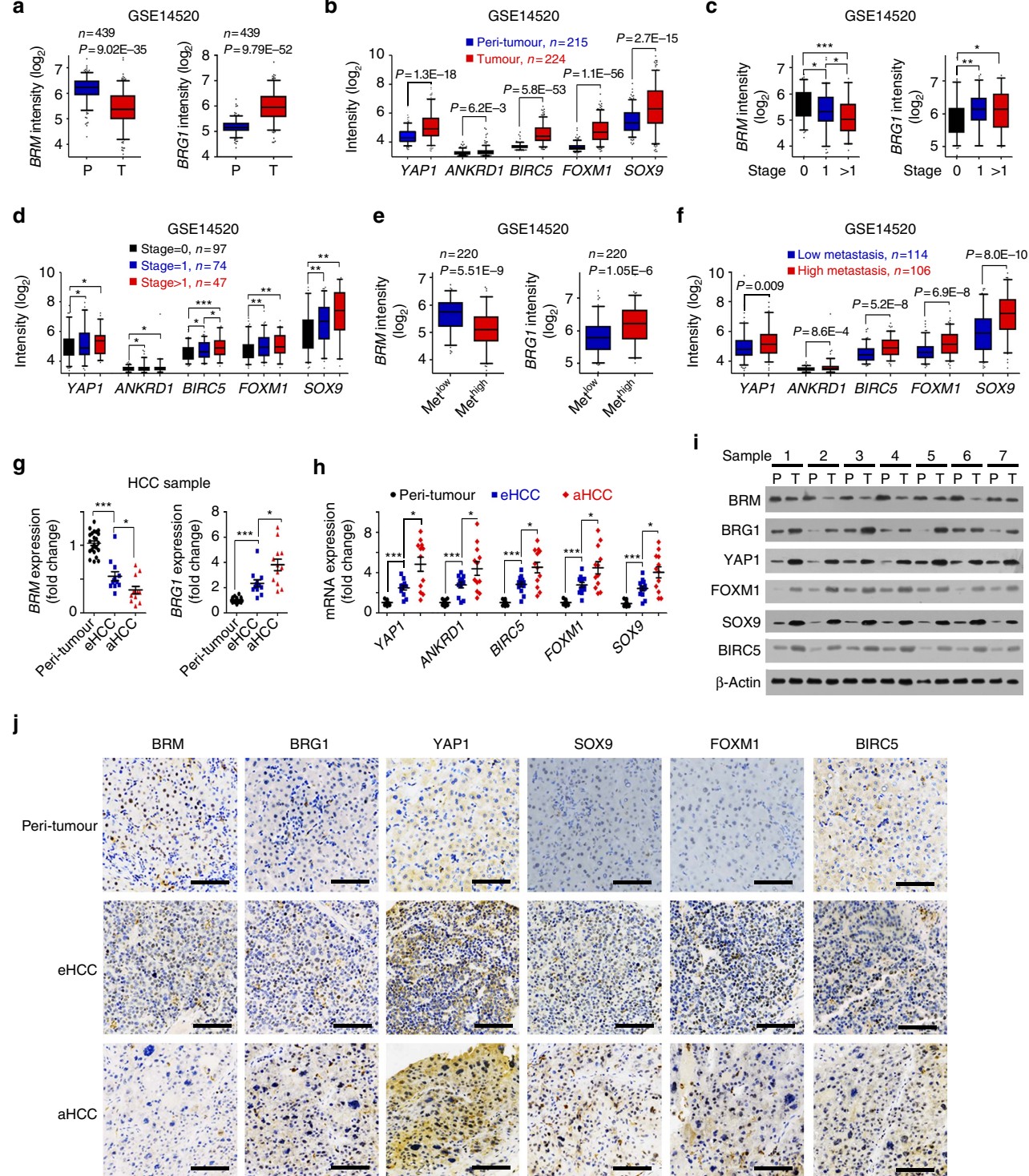

**Figure 7 | Expression levels of BRG1 and YAP1 targets are positively correlated with severity and prognosis of HCC patients.** (**a**) Decreased *BRM* expression (left panels) and increased *BRG1* expression (right panels) in HCC tumour tissues derived from Wang's cohort (GSE14520). R language was used for gene expression analysis. P, peri-tumour; T, tumour. (**b**) Expression levels of *YAP1* and its target genes in peri-tumour and tumour tissues were analysed using R language and Bioconductor, and shown as box and whisker plot. (**c**) Increased *BRG1* expression (left panel) and decreased *BRM* expression (right panel) in advanced HCC patients according to Wang's cohort (GSE14520). (**d**) High expression levels of *YAP1* and its target genes in high metastasis samples derived from Wang's cohort (GSE14520). (**e**,**f**) Decreased *BRM*, increased *BRG1* (**e**) and elevated YAP1 target genes (**f**) exhibited in high metastasis HCC patients according to Wang's cohort (GSE14520). (**g**) Expression levels of *BRG1* and *BRM* were examined in HCC primary samples using RT–qPCR. aHCC, advanced HCC patients; eHCC, early HCC patients; P, peri-tumour. (**h**) Expression levels of *YAP1* and its target genes were examined in HCC primary samples. Twenty-four peri-tumour, 12 eHCC and 12 aHCC were analysed. (**i**,**j**) HCC primary samples were lyzed for immunoblotting (**i**) and immunohistochemical staining (**j**). Scale bars, 100 μm. For **a**–**f**, data are shown as box and whisker plot. Box: interquartile range (IQR); horizontal line within box: median; whiskers: 5–95 percentile. For **g**,**h**, data are shown as means ± s.d. Two tailed Student's *t*-test was used for statistical analysis; *$P < 0.05$, **$P < 0.01$ and ***$P < 0.001$. Data are representative of at least three independent experiments.

expressed in CD90$^+$, EpCAM$^+$, CD24$^+$ or CD44$^+$ CSCs isolated from HCC primary samples (Supplementary Fig. 7D). In addition, *lncBRM* silencing in CD90$^+$, EpCAM$^+$, CD24$^+$ or CD44$^+$ CSCs indeed impaired CSC self-renewal and oncosphere formation (Supplementary Fig. 7E). These observations were consistent with those of CD13$^+$CD133$^+$ cells[11]. In our tested liver CSCs, *lncBRM* was highly expressed in these CSC cells and required for their self-renewal maintenance. In addition, we observed that the ratios of CD13$^+$CD133$^+$ populations in HCC primary samples were around 3~12% and over 90% population percentage in HCC patients (Supplementary Fig. 7F). However, there was no significant relationship between the *lncBRM* expression levels and the ratios of CD13$^+$CD133$^+$ populations. Moreover, *lncBRM* deletion failed to affect asymmetric division ratios and differentiation tendency (Supplementary Fig. 7G,H), suggesting that *lncBRM* mainly regulates the self-renewal of liver CSCs through *lncBRM*-mediated YAP1 activation.

Chromatin structure is modulated by two general classes of complexes: those that covalently modify histone tails and those that remodel nucleosomes in an ATP-dependent manner[27,39]. SWI/SNF remodelling complexes use the energy of ATP hydrolysis to remodel nucleosomes and to regulate gene transcription. Of note, inactivating mutations of several SWI/SNF subunits have been found in various human tumours[26,40,41]; thus, the SWI/SNF complexes have been considered to be tumour suppressors. However, the role of SWI/SNF complexes in cancer remains controversial. In mammals, BAF-containing complexes were termed BAF complexes, which encompass one of two mutually exclusive catalytic ATPase subunits, either BRM (also known as SMARCA2) or BRG1 (also called SMARCA4). Although BRM may have some redundancy with BRG1, these two types of complexes could be largely distinct[42]. BRG1-deficient mice die in early embryonic development, whereas BRM-deficient mice are live and heavier than normal mice[43,44]. BRG1 tends to interact with Zinc finger proteins through its unique amino-terminal domain, whereas BRM associates with proteins containing two ankyrin repeats[45]. Herein we showed that BRG1 promotes the self-renewal of liver CSCs, but BRM exerts an opposite effect (data not shown). Mechanistically, *lncBRM* binds to BRM that is replaced by its homologue BRG1 to form the BRG1-biased BAF complex, which triggers the activation of YAP1 signalling. BRM binding with *lncBRM* may undergo conformational changes to abolish its binding capacity to the BAF complex. Another possibility is that *lncBRM* may bind to the domain of BRM that is critical for the assembly of BRM-biased BAF complex. In fact, increased BRG1 with decreased BRM appears in HCC tumours and this trend is related to severity of HCC patients. We show that the BRG1/BRM switch plays a critical role in the regulation of liver CSC self-renewal and HCC oncogenesis. A recent report showed that the interaction of Yki/Yap1 and Brm, the *Drosophila* homologue of mammalian BRG1, promotes Yki-dependent transcription and tissue growth in *Drosophila*[46]. Here we demonstrate the specific interaction of KLF4 and BRG1 (but not BRM) in liver CSCs to initiate YAP1 signalling activation.

Physical properties of the extracellular matrix (ECM) and mechanical forces are integral to defining tissue architecture and driving specific cell differentiation programmes in embryonic development[47]. In adulthood, tissue homeostasis remains dependent on physical cues, such that perturbations of ECM stiffness are causal to pathological conditions of multiple organs, contributing to ageing and malignant transformation[48]. Mechanotransduction enables cells to perceive and adapt to external forces and physical constraints. YAP1 and Taz can perceive mechanical signals exerted by ECM rigidity and cell shape as mechanotransducers[49], which requires Rho GTPase activity and reorganization of cytoskeleton, but is independent of the Hippo/Lats cascade. We found that Lats1 is undetectable and in liver CSCs following mechanical train (data not shown). Moreover, we noticed that BRG1 KO or BRM KO did not have an impact on phosphorylation signals of MST1/2 and LATS1 in liver CSCs. Therefore, *lncBRM*-mediated YAP1 signalling is independent of the Hippo/Lats cascade in the progression of HCC. As for tumorigenesis, inflammation, changes of microenvironments and stiffness of ECM confine the cell's adhesive area and result in mechanical strain[21]. We added Matrigel, collagen I or methyl-cellulose in sphere formation media followed by oncosphere formation assays. Addition of these three reagents dramatically promoted HCC primary oncosphere formation and augmented expression of pluripotency factors (data not shown), suggesting mechanical strain enhances the initiation of liver CSC self-renewal. However, how mechanical strain modulates the activation of *lncBRM*-mediated YAP1 signalling in liver CSCs needs to be further investigated.

In summary, *lncBRM* is highly expressed in HCC tumours and liver CSCs, which triggers YAP1 signalling to promote self-renewal of liver CSCs and initiate tumour propagation (Supplementary Fig. 7I). Therefore, *lncBRM*, along with YAP1 signalling targets might be used in diagnosis and prognosis, as well as in the development of novel therapeutic drugs against HCC.

## Methods

**Cells lines.** HCC cell lines Huh7, Hep3B and PLC were obtained from Dr Zeguang Han (Shanghai Jiaotong University School of Medicine, Shanghai, China). HCC cell lines were maintained in DMEM medium supplemented with 10% fetal bovine serum, 100 µg ml$^{-1}$ penicillin and 100 U ml$^{-1}$ streptomycin. All cell lines were verified by PCR (*SV40gp6* for 293 T cells; *HBVgp2* for PLC cells; *AFP*, *ALB*, *HBVgp2* and *A2M* for Hep3B; and *AFP* for Huh7 cells). These cell lines were not contaminated by mycoplasma.

For preparation of HCC primary cells, HCC samples were immediately obtained after resection. Tumour bulk was cut into 1 mm$^3$ with scissors, followed by 40 min digestion at 37 °C with collagenase IV (0.05% collagenase IV, 0.05% proteinase, 0.01% DNase and 5 mM CaCl$_2$ in HBS), with shaking every 10 min. Next, the medium were passed through a 70 µm cell strainer and centrifuged at 50 g for 1 min. Supernatant fractions were collected and further centrifuged at 150 g for 8 min and HCC cells were enriched in pellets. After treatment with Red Blood Cell Lysis Buffer for red cell elimination, the HCC primary cells were obtained, followed by FACS, sphere formation and other experiments.

**Antibodies and regents.** Anti-β-actin (catalogue number A1978, 1:5,000), anti-Flag (catalogue number F1804, 1:5,000) and anti-YAP1 (catalogue number Y4770, 1:500) antibodies were purchased from Sigma-Aldrich. Anti-BRG1 (catalogue number 21634-1-AP, 1:500), anti-FOXM1 (catalogue number 13147-1-AP, 1:1,000) and anti-KLF4 (catalogue number 11880-1-AP, 1:500) antibodies were obtained from Proteintech Group. Anti-BRM (catalogue number 11966, 1:1,000), anti-BAF170 (catalogue number 12760, 1:1,000), anti-SNF5 (catalogue number 8745, 1:1,000), anti-ARID1A (catalogue number 12354, 1:1,000), anti-Baf155 (catalogue number 11956, 1:1,000), anti-Histone H3 (catalog 4499, 1:2,000) and anti-Oct4 (catalogue number 2750, 1:5,000) antibodies were from Cell signalling technology. Anti-SOX9 antibody (catalogue number AB5535, 1:500) was obtained from Millipore. Anti-BIRC5 antibody (catalogue number sc-8807, 1:1,000) was purchased from Santa Cruz Biotechnology. Horseradish peroxidase (HRP)-conjugated secondary antibodies (catalogue number sc-2004, sc-2005, 1:500) were from Santa Cruz Biotechnology. Phycoerythrin-conjugated CD133 antibody (catalogue number 130-098-826, 1:200) was from Miltenyi Biotec. Fluorescein isothiocyanate-conjugated CD13 antibody (catalogue number 11-0138, 1:500) was from eBioscience. Alexa488-conjugated donkey anti-mouse IgG (catalogue number R37114, 1:500), Alexa594-conjugated donkey anti-rabbit IgG (catalogue number R37119, 1:500) and Alexa594-conjugated donkey anti-mouse IgG (catalogue number R37115, 1:500) antibodies were from Molecular Probes Life Technologies. Epidermal growth factor (catalogue number E5036-200UG), PEG5000 (catalogue number 175233-46-2) and 4,6-diamidino-2-phenylindole (catalogue number 28718-90-3) were purchased from Sigma-Aldrich; N2 supplement (catalogue number 17502-048) and B27 (catalogue number 17504-044) were from Life Technologies; basic fibroblast growth factor (catalogue number GF446-50UG) was from Millpore. LightShift Chemiluminescent RNA EMSA Kit (catalogue number 20158) and Chemiluminescent Nucleic Acid Detection Module (catalogue number 89880) were from Thermo Scientific.

T7 RNA polymerase (catalogue number 10881767001) and Biotin RNA Labeling Mix (catalogue number 11685597910) were from Roche.

**EMSA and RNA EMSA assays.** BRG1 and KLF4 EMSA assays were performed using standardized EMSA procedure, and KLF4 and BRG1 were obtained using immunoprecipitation from oncospheres[50]. Biotin-labelled *YAP1* promoter ($-420 \sim -380$) was purchased from Sangon Company (Shanghai, China) and labelled with [r-$^{32}$P] dATP according to standard protocol. Probes and precipitated proteins were incubated in EMSA binding buffer and mobility shift assay was performed using gel electrophoresis.

For RNA EMSA, human BRM plasmid was transfected into Huh7 cells, then cell nuclear extracts were isolated from BRM-overexpressed oncospheres. LncBRM-specific probes were labelled with Biotin using Biotin RNA Labeling Mix (Roche). BRM nuclear extracts and biotin-labelled probe were incubated in $1 \times$ REMSA binding buffer supplementing with Glycerol, transfer RNA and dithiothreitol for 30 min according to LightShift Chemiluminescent RNA EMSA Kit (Thermo Scientific). Native PAGE was performed for separating components followed by transferring onto positively charged NC film (Beyotime Biotechnology). After ultraviolet cross-linking, HRP-conjugated streptavidin was added to detect the biotin signalling according to Chemiluminescent Nucleic Acid Detection Module (Thermo Scientific).

**RNA pulldown assay.** For RNA pulldown assays, biotin-labelled *lncBRM* transcripts, *lncBRM* intron sequence and its antisense were obtained using *in vitro* transcription with T7 RNA polymerase and Biotin RNA Labeling Mix (Roche), followed by 4 h incubation at 37 °C with oncosphere cell lysates. Then streptavidin-conjugated agarose beads were used for centrifugal enrichment. Precipitated components were separated using SDS–PAGE, followed by silver staining[51]. Differential bands were cut for mass spectrometry (LTQ Orbitrap XL).

**RNA immunoprecipitation.** Primary spheres were treated with 1% formaldehyde for 15 min, then dissolved with modified RIPA buffer (150 mM NaCl, 0.5% sodium deoxycholate, 0.2% SDS, 1% NP40, 1%Triton X-100, 1 mM EDTA and 50 mM Tris pH 8.0), supplementing with Protector RNase inhibitor and protease inhibitors (Roche). Samples were sonicated three times on ice, followed by $13,800 g$ centrifugation for 10 min. Supernatants were incubated with protein A/G beads for 1 h, followed by 4 h incubation with indicted antibodies and subsequent 2 h incubation with protein A/G beads. *LncBRM* enrichment was examined using RT–qPCR, IgG enrichment served as controls[52]. Primer sequences are shown in the Supplementary Table 3.

**Northern blotting.** Total RNA was extracted from HCC samples or oncospheres using standard TRIZOL methods, followed by electrophoresis with formaldehyde denaturing agarose gel. Samples were transferred to positively charged NC film (Beyotime Biotechnology) using $20 \times$ SSC buffer (3.0 M NaCl, 0.3 M sodium citrate, pH 7.0). After ultraviolet cross-linking, membrane was incubated with hybrid buffer for 2 h prehybridization, followed by incubation with Biotin-labelled RNA probes generated by *in vitro* transcription at 65 °C for 20 h. Biotin signals were detected with HRP-conjugated streptavidin according to the introduction of Chemiluminescent Nucleic Acid Detection Module (Thermo Scientific).

**RNA–FISH assay.** Fluorescence-conjugated *lncBRM* probes for RNA–FISH were generated according to protocols of Biosearch Technologies. HCC samples or spheres were treated in a non-denaturing condition, followed by hybridization with DNA probe sets. Then antibodies were added after RNA hybridization for co-localization of RNA and indicated proteins. All experiments were performed as described in manuals of Biosearch Technologies[11]. Treated samples were visualized by confocal microscopy (FV1000, Olympus).

**Lentivirus generation and infection.** For *lncBRM* and other lncRNA knockdown, we constructed pSiCoR shRNA system. Sequences of shRNAs were designed according to online tools of Clontech Company and cloned into pSiCoR vector. For virus generation, we transfected 293T cells with pSiCoR along with package plasmids (4 mg pSiCoR vector, 1 mg VSVG, 1 mg pMDL g/p RRE and 2 mg RSV-REV were used for 10 cm dish). Hep3B, Huh7 and HCC primary cells were infected by virus supernatants or PEG5000 (Sigma)-enriched precipitates. After purified with puro or green fluorescent protein (GFP), stable cell lines were established. For overexpression, similar strategy was used. shRNA sequences are listed in Supplementary Table 4.

**Western blotting.** Primary cells and spheres were crushed with RIPA buffer (150 mM NaCl, 0.5% sodium deoxycholate, 0.1% SDS, 1% NP40, 1 mM EDTA and 50 mM Tris pH 8.0), followed by separation with SDS–PAGE. Then samples were transferred to NC membrane and incubated with primary antibody in 5% milk. After washing with TBST three times, membranes were blotted with HRP-conjugated secondary antibodies for visualization[53]. The uncropped blots were shown in Supplementary Fig. 8.

**Sphere formation assay.** One thousand PLC or Hep3B cells were grown in sphere formation medium (DMEM supplemented with 20 ng ml$^{-1}$ basic fibroblast growth factor, 20 ng ml$^{-1}$ epidermal growth factor, N2 and B27). Two weeks later, spheres larger than 100 μm were counted and photographs were taken. For HCC samples, 5,000 primary cells were used for sphere formation.

**Diluted xenograft tumour formation.** For tumour propagation analysis, *lncBRM* or YAP1 silenced tumour cells ($1 \times 10^6$) were subcutaneously injected into BALB/c mice. Tumour volumes were measured every 4 days. For tumour-initiating capacity assay, 10, $10^2$, $10^3$, $10^4$ and $10^5$ cells were injected into BALB/c nude mice, respectively. Three months later, tumour formation was counted, followed by calculation of ratios of tumour-free mice and tumour-initiating cells[9].

**ChIP immunoblotting assay.** ChIP immunoblotting assay and size fractionation were performed using oncospheres[25]. For ChIP assays, oncospheres were treated with 1% formaldehyde for 10 min for crosslinking ( + crosslink), crashed with SDS lysis buffer and followed by ultrasonication. Standard ChIP assay was performed using KLF4 antibody. The eluate (500 μl) was layered onto 35 ml 5–30% (V/V) glycerol gradients followed by ultracentrifugation. S Beckman SW28 rotor was used for ultracentrifugation at $55,200 g$ for 30 h. Then, fractions were carefully collected and the elution gradients were concentrated to 50 μl using centrifugal concentration tubes. Finally, the samples were analysed using western blotting and PCR assays. For BRG1/BRM switch, spheres were crushed with RIPA buffer, followed by glycerol gradient ultracentrifugation.

**CRISPR/Cas9 KO system.** *BRG1*, *BRM*, *KLF4*, *YAP1* and *YAP1* promoter-deficient and *YAP1* promoter mutant HCC primary cells were established using a CRISPR/Cas9 system[8,25]. Briefly, single guide RNA (sgRNA) was designed by online CRISPR Design Tool (http://tools.genome-engineering.org) and cloned into lenti Cas9-EGFPvector (Addgene catalogue number 63592). After confirming the cutting efficiency of sgRNA, lenti Cas9 was transfected in 293T cells with pVSVg (Addgene catalogue number 8454) and psPAX2 (Addgene catalogue number 12260) for 48 h. Supernatants were collected and concentrated with PEG5000 (Sigma Aldrich), then infected primary cells for 5 days. GFP-positive cells (usually >80%) were enriched, followed by KO efficiency examination using western blotting. For *YAP1* promoter KO, a pair of sgRNAs that were derived from the left locus and the right locus of BRG1/KLF4-binding region on *YAP1* promoter were cloned into puro lentiCRISPRv2 and GFP lentiCRISPRv2. Two a lentivirus were co-transfected and the KO efficiency was confirmed by DNA sequencing. For *YAP1* promoter mutation, sgRNA and template DNA were transfected into Huh7 cells, followed by selection and monoclonalization. Genome DNA was extracted and *YAP1* promoter established clones were examined by T7 endonuclease I cleavage, and confirmed by DNA sequencing.

**Statistical methods.** R language and Bioconductor were used to analyse online available data sets and results were shown as box and whisker plot[54]. For CSC ratio analysis, extreme limiting dilution analysis was performed using tumour-free mouse numbers of 10, $10^2$, $10^3$, $10^4$ and $10^5$ cells[55]. For most trials, two tailed Student's *t*-test was used for statistical analysis. $P < 0.05$ was considered as statistically significant.

**Study approval.** Human HCC samples were obtained from the Department of Hepatobiliary Surgery, PLA General Hospital (Beijing, China) with informed consents from all human participants, according to the Institutional Review Board approval. We numbered HCC primary samples according to receiving date and used the samples without artificial bias. Six-week-old female BALB/c nude mice were purchased from the Animal Center of the Chinese Academy of Medical Sciences (Beijing, China). All experiments involving mice were approved by the institutional committee of Institute of Biophysics, Chinese Academy of Sciences. Studies were conducted in a blinded manner. Animals were randomly assigned to groups. No sample-size estimates were used and no animals or samples were excluded from analyses.

**Data availability.** The microarray data used in the study (GSE66529, GSE14520, GSE25097 and GSE54238) are available in a public repository from EBI (http://www.ebi.ac.uk/) or NCBI (http://www.ncbi.nlm.nih.gov/gds/?term). All relevant data are available from the authors.

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

## Acknowledgements

We thank Junying Jia for technical support. We thank Jing Li (Cnkingbio Company Ltd, Beijing, China) for technical support. This work was supported by the National Natural Science Foundation of China (91419308, 91640203, 81330047, 31530093, 81272270, 81402459 and 81101531); 973 Program of the MOST of China (2015CB553705); the State Projects of Essential Drug Research and development (2012ZX09103301-041); the Strategic Priority Research Programs of the Chinese Academy of Sciences (XDA01010407 and XDA01020203); Postdoctoral innovation programme of China Postdoctoral Science Foundation.

## Author contributions

P.Z. designed and performed experiments, analysed the data and wrote the paper. Y.W. performed experiments and analysed data. J.W., G.H., B.L., B.Y., Y.D., S.W., G.G. and Y.T. analysed the data. L.H. provided HCC samples and analysed the data. Z.F. initiated the study, organized, designed and wrote the paper.

## Additional information

**Competing financial interests:** The authors declare no competing financial interests.

