## [Peer Review File · Nature Communications]

Reviewer #1 (Remarks to the Author): Expert in liver cancer and cancer stem cells

NCOMMS-16-06371 by Pingping Zhu et al
 In their recent study, the authors have used array technologies to identify differentially expressed lncRNAs between CD13+ CD133+ (referred to as liver CSCs) and CD13- CD133- (referred to as non-CSCs) cell populations from several HCC cell lines. They have identified lncTCF7 to contribute to self-renewal of liver CSCs (Wang et al, Cell Stem Cell 2015). In the current study, they extended this study by characterizing the top 10 differentially expressed lncRNAs including lncTCF7 as a control. They found lncBrm as another lncRNA to contribute to self-renewal of liver CSCs through binding to BAF complex thereby activating Yap1. They also found that lncBrm expression is correlated with tumor severity of HCC patients. They suggest that lncBrm and Yap1 signaling may serve as new biomarkers and potential drug targets for HCC. This is a comprehensive study that contains 7 multi-paneled figures and 6 multi-paneled supplemental figures. The manuscript is well written. Authors are commended for their heroic effort. While some of the panels are too small to read, generally, most of the figures are of good quality. While many of the experiments are well executed, the main concern I have is a lack of conceptual novelty for overall study design and execution. Some of the experiments also require additional controls. Here are some of my main concerns.

1) Lack of conceptual novelty: There are a huge similarity between this study and their Cell Stem Cell paper in regarding to experimental approaches, design and layouts. The main point of the current study is to show the role of another lncRNA (lncBrm) in CSC self-renewal. The main conclusion is based on shRNA to knockdown lncBrm. While this approach has an inherited issue (see #2 below), the finding does not add much more to our understanding of liver CSC. The key question in my mind should be to demonstrate if lncBrm regulates the maintenance of liver CSC. For example, does it regulate asymmetric cell division, differentiation in addition to self-renewal? Is it all through the Yap1 signaling? It should be noted that the role of YAP1 in HCC is well established. An interesting question remains as whether lncBrm functionally interacts with lncTCF7 since both of them act on the same phenotype.

2) Problems with shRNA to study lncRNA: The main functional studies of this paper rely heavily on the use of shRNA to knockdown lncBrm. Considering possible off-target activities of shRNA, especially when using to characterize a completely uncharacterized lncRNA, i.e, LINCR-0003 or lncBrm, a better study design should be considered. For example, it is necessary to use 2-3 different siRNAs to show that similar degree of knockdown gives you similar phenotype. Using a similar strategy in Buehler E et al, PLOS ONE 7: e51942, 2012, design a 9-11 siRNA control to show that the 9-11 control doesn't knockdown the targeted lncRNA and has no phenotype. These are general rules for targeting mRNA and possibly lncRNA. For lncRNA, additional controls should be also considered such as the use of CRISPR to delete this lncRNA or anti-sense oligos to knockdown the targeted lncRNA especially when it is a nuclear lncRNA like lncBrm. For the current study, the data on shRNA to knockdown lncBrm-specific signaling are not convincing since lncBrm is mainly nuclear while RNAi mechanism occurs mainly in the cytoplasm. Without these additional controls, the link of lncBrm to liver CSC is not convincing.

3) Lack of careful planning and consistency: clinical samples are poorly described. Considering heterogeneity in biospecimens, it is important to know about histological characteristics, clinical stages, etc. When assessing figures containing clinical specimens, it is noted that different cases are used for different experiments. This is similar to the metaphor of apples and oranges. In addition, in Fig 1E, it is important to assess the variability of baseline levels of lncBrm. I suggest normalizing all lncBrm readings to one non-tumor sample that has a lowest expression.

4) The key question of this study and that of their Cell Stem Cell paper is how lncRNAs regulate the self-renewal of liver CSCs. Many different types of liver CSCs have been described by the expression of different CSC markers such as CD90, EpCAM, CD24, CD44, Sal4, etc. It will be interesting to determine if lncBrm regulates these liver CSCs.

Reviewer #2 (Remarks to the Author): Expert in Brm/Brg BAF complex

The manuscript by Zhu et al reports on the discovery of a novel long non coding RNA (lncRNA) highly expressed in liver cancer stem cells, that can interact with the BRM catalytic subunit of the SWI/SNF complex. Binding of lncBrm to the BRM protein sequesters it from the SWI/SNF complex, causing a switch from a BRM-containing SWI/SNF to a BRG1-containing SWI/SNF complex. This switch causes expression of the YAP-1 transcription factor and its downstream targets, leading to cancer stem cells with higher self-renewal capacity. The authors further demonstrate that ablation of either lncBrm or YAP-1 diminishes the renewal capacity and the in vivo tumorigenic capacity of liver cancer stem cells. Moreover, increased expression of BRG1 and YAP1 are correlated with advanced disease, increased metastasis and reduced survival in hepatocellular carcinoma human patients.

The work presented by Zhu et al is of notably high quality. The data presented is robust, includes both loss-of-function and gain-of-function experiments throughout, each producing the predicted effect (eg knockdown of lncBRM impairs oncosphere formation while over-expression of lncBRM enhances oncosphere formation), and the experiments are generally extremely thorough including pursuit of mechanistic insight. The data presented is strongly supportive of the authors' conclusions. Since hepatocellular carcinoma is one of the most common types of cancer and relapsed disease is very difficult to treat, the work also carries substantial impact. I have a few queries which are in general relatively minor or seek clarification:

1. With respect to Figure 3G, the manuscript states "Notably, we observed that Brm-embedded BAF complex was dramatically declined in oncosphere cells (Figure 3G)," If I understand the data in 3G correctly, it shows that indeed IP of BAF170 from shows lower levels of BRM in spheres than non-spheres. However, in that same panel, IP of ARID1A (another SWI/SNF subunit) shows precisely the opposite. The authors need to explain this.
2. It is initially surprising in 3B that IP of lncBRM seems to show enrichment of BRM given that BRM typically exists tightly bound to other members of the SWI/SNF complex. And yet this is conceivable if the authors' conclusion is correct that lncBRM functions to sequester BRM away from the SWI/SNF complex to facilitate a switch to BRG1. It would be informative if the authors would comment on whether they looked for other SWI/SNF subunits following IP of lncBRM.
3. The authors should provide a direct statement with reference that the CD133 fraction of cells has been shown to be the stem cell population in this model.
4. The authors cite use of "Wang's cohort (GSE1452031)" as being supportive of their data/model. Were other similar cohorts analyzed that did not support the model? Or was this the only relevant cohort? This should be clarified.
5. BRG1 and BRM are quite similar and some antibodies cross-react and recognize both. The authors should provide a specific statement that the antibodies that they are using are specific and recognize only BRG1 or BRM without cross-reacting to the other. Reference to a commercial product insert is fine if that covers this, or even better if the authors have confirmed this themselves, although confirmation by the authors is not essential if covered by the commercial insert.

Minor points

1. Since all the samples used are of human origin, the gene labeling should be in capital letters
2. The manuscript is well-written and clear although there are a few grammatical errors that should be corrected.

Reviewer #3 (Remarks to the Author): Expert in lncRNAs

This manuscript submitted by Pingping Zhu and Yanying Wang et al. demonstrates the dramatic genetic role of lncBrm as a modulator of the self-renewal activity in liver cancer stem cells (liver CSCs). lncBrm has been categorized as an uncharacterized non-coding RNA called LINCR-0003 and these authors identifies this lncBrm from their own transcriptome microarray analysis of liver CSCs and non-CSCs. They demonstrate that hepatocellular carcinoma (HCC) and liver CSCs has a high expression of lncBrm and this lncRNA itself has a tumorigenic and oncosphere formation

capacity. Furthermore, they clarify the association between IncBrm and BAF-Yap1 signaling connections which is required for maintenance of the stemness of liver CSCs and tumorigenesis. All of their results and conclusions are based on their abundant and detailed experiments not only in vitro but also in vivo and clinical phase.

For these reason, I would suggest that this manuscript will be acceptable for the journal of Nature Communications after several minor revisions listed below.

Minor revisions,
#1: These authors are using HCC primary cells for sphere formation assay such like a cell line to divide these clinical tumor cells into CSC and non-CSC characters. There are, however, no explanations how to establish these clinical cancer cell lines from HCC primary calls. Please describe detailed information of this point.

#2: I would suggest that if you can put an illustrated schema of the interactions between IncBrm, BAF-complex family and Yap1 signaling pathway, it will be a big help for readers to understand your results well.

Reviewer #4 (Remarks to the Author): Expert in YAP

This paper describes the regulation of HCC stem cells by the long non-coding (lnc) RNA IncBrm and also elucidates the underlying mechanism by showing that IncBrm stimulates the expression of YAP pathway components. This is a novel and interesting finding which is supported by a wealth of data shown in this paper. Given the importance of the YAP pathway in liver cancer and the emerging function of lnc RNAs in cancer, I think this paper is an important contribution that will be influential in the field.

Major comments
I have two main criticisms. First, this paper contains a huge amount of work, but often described in a brevity and superficial way that makes it difficult to understand what exactly has been done and how the authors arrive at their conclusions. My second point is that a critical junction that links the main parts of the paper is hardly borne out. This junction claims that IncBrm regulates the expression of YAP pathway components and YAP signalling. It only is addressed by a rather incomprehensible Fig. 5G and a poor Northern blot (Fig. 5H). This claim should be strengthened, e.g. by showing the effects of IncBrm on the recruitment of Brm and Brg1 to the YAP promoter and the effects on the transcriptional output of the corresponding promoter elements (e.g. luciferase assays).

Specific comments
Fig. 3G should be explained in more detail, especially what complexes the antibodies actually precipitate, and what the inverse ratios between Brm and Brg1 in the Baf170 and ARID1 IPs mean.

The behaviour of Brg1 and Brm is different in the Arid1a IPs between Figs. 3G and H. In Fig. 3G high IncBRM expression in spheres enhances Brm and decrease Brg1 co-IP. In Fig. 3H adding IncBrm has the opposite effect. Why?

Figs. 3G, H and I need loading controls showing the total amount of Brm and Brg1 in lysates.

Figs. 3K and J are so poorly described that they are un-interpretable. Moreover, the description provided seems to contradict the results of previous panels.

What does Fig. 4A show? qPCR?

Fig. 4B. Is only the expression of YAP pathway components altered or also activities? This easily could be addressed by blotting with phosphospecific antibodies for MST1/2, LATS1/2 or YAP1/2.

Fig. 5D. These assays are insufficiently described making it difficult to understand what has been done and what the conclusion was.

Fig. 5H is a critical experiment showing that IncBrm changes YAP1 mRNA expression supporting the main conclusions that IncBrm enhances YAP signalling. Unfortunately, this blot is of poor

quality with air bubbles and a very fuzzy appearance. It should be replaced by a better quality blot.

Fig. 6G needs quantification and results displayed as graph with error bars.

"Verteporfin treatment abolished enhanced sphere formation induced by IncBrm overexpression (Supplementary Figure 5F)." The figure does not contain a condition of IncBrm overexpression, which obviously would be necessary for making such a statement.

The regulation of YAP by Brg1 has been recently reported in Drosophila and should be discussed and referenced. 1

Minor Details about MS identification of BRM should be provided. comments

"DNase I sensibility assays..." should be "DNase I sensitivity assays..."

"In parallel, H3K4me3 antibody did not enriched Yap1 promoter regions in Brg1 KO oncospheres..." It should be explained what this result means.

Legend to Fig. 5. "(A) HCC primary spheres were performed for co-IP using Klf4 antibody." "(D) HCC primary sphere cells were crosslinked with formaldehyde and then performed ChIP using Klf4 antibody, ..." These sentences are not English.

YM155 is not a Birc5 inhibitor, but an inhibitor of Birc5 expression. It also is not specific and may have other effects.2

"Expression levels of Brg1 and Yap1 signaling targets are positively corrected with severity and prognosis of HCC patients" should read "Expression levels of Brg1 and Yap1 signaling targets are positively correlated with severity and prognosis of HCC patients"

P10. "...immunoblotting (Figure 7H)..." should be "...immunoblotting (Figure 7I)..."

P11. "Mechanically, IncBrm binds to Brm..." should be "Mechanistically, IncBrm binds to Brm..."

References

- 1 Zhu, Y. et al. Brahma regulates the Hippo pathway activity through forming complex with Yki-Sd and regulating the transcription of Crumbs. Cellular signalling 27, 606-613, doi:10.1016/j.cellsig.2014.12.002 (2015).
- 2 Rauch, A. et al. Survivin and YM155: how faithful is the liaison? Biochimica et biophysica acta 1845, 202-220, doi:10.1016/j.bbcan.2014.01.003 (2014).

Point-by-point response to reviewers' comments

Reviewer #1

Comment; While the revisions have significantly improved, there are some concerns. I recommend that the concerns be addressed before acceptance. My concerns and suggestions are stated below.

1. Comment 1 Response to Answer: Since the authors claim that this is novel and "quite different", their response and later works only provide a different mechanism that regulates the "same" self-renewal property. While it is likely that you have multiple mechanisms that can drive one phenotype, this by itself provides a "quite different" mechanism in driving self-renewal, but does not provide a "novel" aspect. This also brings up the concept that IncBRM can be used as a biomarker for diagnosis or potential drug targets as commented by the authors. Their data brings up the issue that because many mechanisms are driving this characteristic in the same cell group (CD13+CD133+), targeting one mechanism is not enough and thus, does not serve to be a good therapeutic target. This statement should be modified as their evidence (i.e. in their new data presented on Attached Figure 2A) suggests that the synergistic combination of

lncTCF7+lncBRM together have a better effect than just one, indicating that two lncTCF7 and lncBRM together are better than lncBRM alone or lncTCF7 alone.

Answer: This is a good point. We addressed this issue in the discussion section.

2. Comment 2 Response to Answer: In Attached Figure 1 B, the figure only shows CD133+ cells. Is this the same CD13+CD133+ cells? The authors show two different timepoint data. It's not comparable in terms of analysis. The data changes in CD13-CD133- or CD13+CD133+ cells from shCtrl or shLncRBM for the same date should be shown (i.e. day 5 or day 20 for both control and lncBRM silencing).

Answer: Yes, the CD13⁺CD133⁺ population had the similar differentiation tendency as the CD133⁺ population. For each timepoint, CD13⁺CD133⁺ cells treated with shCtrl or shLncRBM displayed similar differentiation tendency. As controls for CD13⁻CD133⁻ cells, shCtrl or shLncRBM treatment showed similar differentiation ratios to those of their counterpart CD13⁺CD133⁺ cells treated with shCtrl or shLncRBM. We addressed this issue in the discussion section.

3. Comment 3 Response to Answer: Since the authors picked samples with the highest lncBRM, is there also an association with CD13+CD133+ cell populations? One important aspect that is missing is the population of CD13+CD133+ in each samples, possibly being correlated positively with the lncBRM expression. This data will provide the importance of CD13+CD133+ in the lncBRM relationship in clinical aspects and the clinical evidence that CSCs are driven lncBRM.

Answer: For selected lncBRM highly expressing samples, the ratios of the populations of CD13+CD133+ in HCC primary samples were around 3~12%. There was no significant relationship between the lncBRM expression levels and the ratios of CD13+CD133+ populations. We addressed this issue in the discussion section.

4. Comment 4 Response to Answer: The authors provide nice data demonstrating that lncBRM silencing reduces spheroid formation in CD90+, EPCAM+, CD24+, or CD44+ cells in Attached Figure 3. Are all these from HCC cells or the samples? This is not stated. The data is interesting in that you have one mechanism which is specific to CD13+CD133+ that seems to regulate the self-renewal mechanism in four other CSCs. This brings up the concern of whether this lncBRM has a specific effect on CD13+CD133+ or in general. What about the CD90-, EPCAM-, CD24-, CD44- cells? This is an important control and brings up a point that they used all cells for some of their studies (Hep3b and Huh7). They did not perform FACs to sort out the true population for some of their studies as their main focus is on this CSCs and yet their original findings are the CD13+CD133+ cells. Are they now arguing that lncBRM targets ALL CSCs? What is the population % of CD13+CD133+, are they targeting ONLY those or everything? This fluctuation is a concern. What is lncBRM regulating the self-renewal of, all CSCs? Or CD13+CD133+?

Answer: For the Attached Figure 3, CD90⁺, EPCAM⁺, CD24⁺, or CD44⁺ cells were isolated from HCC primary samples for these experiments. Moreover, lncBRM was also highly expressed in these CD90⁺, EPCAM⁺, CD24⁺, and CD44⁺ cells. However, lncBRM was almost undetectable in the CD90⁻, EPCAM⁻, CD24⁻, and CD44⁻ cells, which was in agreement with the observations of CD13⁻CD133⁻ cells. In addition, lncBRM depletion the CD90⁻, EPCAM⁻, CD24⁻, or CD44⁻ cells did not affect spheroid formation compared to shCtrl treated respective cells. We tested over 50 HCC primary samples; the population percentage of CD13⁺CD133⁺ cells was over 90%. In sum, in our tested liver CSCs, lncBRM was highly expressed in these CSC cells and required for their self-renewal maintenance. We discussed this issue in the discussion section.

REVIEWERS' COMMENTS:

Reviewer #1 (Remarks to the Author):

Overall the paper has improved significantly. However, where is the data that the authors discussed about in all the comments? These data are important to the authors conclusion that

It should be noted that if the authors are going to make comments and state data in their discussion, they should at least show the data, make note of where the data came from or cite. Otherwise, a discussion is not enough.

Comment 2: response to answer: Comment: The authors use this data to conclude in their paper that IncBRM mainly affects self-renewal and not differentiation (as commented on page 13 line 344) and yet they removed the data.

Comment 4: Response to answer: Comment: The authors removed the data in which this comment was made and had discussed this issue in the Discussion instead. These data were used to conclude that IncBRM was highly expressed in CSCs and required for self-renewal. What happened to these data? Where is the new data (i.e. CD13+CD133+ population was over 90%, IncBRM expression in CD90+, etc. cells)? Discussion is worth note, but there is no data or citations to support the authors comment on Page 12 line 334-341.

These two comments are important for their conclusions. The data needs to be there.

Reviewer #1:

Overall the paper has improved significantly. However, where is the data that the authors discussed about in all the comments? These data are important to the authors' conclusion that It should be noted that if the authors are going to make comments and state data in their discussion, they should at least show the data, make note of where the data came from or cite. Otherwise, a discussion is not enough.

Answer: This is a good suggestion. Since one of the reviewers commented that we put lots of data in our paper, we thought that we could not put additional data in our revised version. We thus provided these data in the attached figures to address the reviewer's comments. For the previous revision version, we only addressed the reviewer's comments in the discussion section without providing these respective data. As suggested, we provided these data in in the new Supplementary Figure 7D-H. We added these figures in the corresponding sentences of the discussion section.

Comment 2: response to answer: Comment: The authors use this data to conclude in their paper that IncBRM mainly affects self-renewal and not differentiation (as commented on page 13 line 344) and yet they removed the data.

Answer: We provided these results in the new Supplementary Figure 7G, H and added these figures in the according line.

Comment 4: Response to answer: Comment: The authors removed the data in which this comment was made and had discussed this issue in the Discussion instead. These data were used to conclude that IncBRM was highly expressed in CSCs and required for self-renewal. What happened to these data? Where is the new data (i.e. CD13+CD133+ population was over 90%, IncBRM expression in CD90+, etc. cells)? Discussion is worth note, but there is no data or citations to support the authors comment on Page 12 line 334-341.

Answer: We provided these results in the new Supplementary Figure 7D-F and added these figures in the according line.